# The Relationship between Leadership and Performance in Enhancing the Sustainability of Social Enterprises

## Mugoan Jeong

College of Business, Kyungnam University, Changwon-si 51767, Republic of Korea; duryjmg@kyungnam.ac.kr; Tel.: +82-55-249-2487

**Abstract:** As social enterprises are established for the purpose of solving local problems, the sustainability of social enterprises is also important for local development. In order to increase the sustainability of social enterprises, performance improvement is necessary, and research on finding leadership styles suitable for social enterprises continues to be an important method of performance improvement. However, despite considerable interest in social enterprises over the past few decades, no consistent conclusion has been reached about what leadership style is appropriate for social enterprises. The present study aimed to find a suitable leadership style for social enterprises within the major leadership styles recognized in commercial enterprises. This study investigated the impact of the three major leadership styles in commercial enterprises on satisfaction and performance in social enterprises. Based on prior research, transformational leadership, with its capacity to promote change and support the achievement of goals, was considered appropriate for social enterprises, and entrepreneurship was also considered as challenge is important for balancing economic and social purposes. Servant leadership also focused on the importance of providing support for employees on the basis that employees' success is the success of social enterprises. In this study, in order to broadly understand the performance of different leadership styles, job satisfaction and leader satisfaction were considered at the individual level, and economic performance and social performance were considered at the firm level. To support the argument of this study, the researcher aimed to survey employees who perceive a leader as possessing leadership qualities in social enterprises. For this purpose, a survey on transaction leadership was conducted among employees who participated in a three-week training session related to social enterprises. And on the last day of training, three leadership and four outcome variables were surveyed. Analyzing data form 170 respondents in 52 social enterprises whose evaluation of their leader's transactional leadership was higher than the average, the present study found that transformational leadership had a positive relationship with all four variables considered as leadership outcomes. And this study found evidence that entrepreneurship had a positive relationship with job and leader satisfactions, but the impact of entrepreneurship on economic and social performance was insignificant. However, servant leadership had a positive relationship with leader satisfaction and social performance, but the impact of servant leadership on job satisfaction and economic performance was insignificant. These results contribute to confirming that the emphasis on leadership styles in commercial enterprises can effectively operate in social enterprises as well, and that the effectiveness of leadership can vary depending on the desired outcomes.

**Keywords:** transformational leadership; entrepreneurship; servant leadership; social enterprises



## 1. Introduction

The social enterprise, regardless of the level of national development, shares a globally consistent founding purpose in seeking to address socially significant issues through social means [1]. However, since social concerns vary by region, social enterprises can pursue different objectives reflecting the characteristics of the region [2,3]. Ghauri et al. [4] argued that social enterprises are powerful tools for alleviating poverty. Ansari et al. [5] insisted

that they are important means of promoting integrated growth in the portion of the labor market receiving minimum wage. Despite numerous studies on social enterprises over the past several decades, neither a systemic concept nor accumulated knowledge of social enterprises has been provided [6,7]. Stevens et al. [8] believed that the nature of social enterprises, which seek both social and economic value simultaneously, suggests that the definition of these values can vary. Consequently, they argued that the success factors of social enterprises could change depending on how their values are perceived. These researchers argued that social enterprises do not universally interpret or provide clear goals or directions to everyone, irrespective of observers. Instead, they focus on the diversity of interpretations that can arise from the same word, depending on the viewer or the time and era. So, they insist that such diversity is inherent. Even to the extent of mentioning that the increase in diversity was challenging the creation of knowledge based on scientific evidence, Smith and Steven [7] considered it difficult to produce a consistent discussion or conclusion regarding social enterprises. However, such arguments overlooked the fact that social enterprises constitute another type of organization that can provide a systematic concept of, or accumulated knowledge about, social enterprises. Most of all, considering the existence of the clear designation 'social enterprise' and the fact that the research targeting them has been some of the most vigorously conducted at the organizational level in recent decades, their arguments were sufficiently irrational [1,7,9].

The argument that researches on social enterprises find it challenging to provide consistent conclusions or systematic concepts is also evident in studies on leadership within social enterprises. Research on suitable leadership for social enterprises has similarly presented various claims, making it difficult to reach a consistent conclusion. Some researchers [7,10] have observed that social enterprises have a distinctive leadership style—so-called social entrepreneurship—stemming from the fact that social enterprises are different from commercial enterprises. Their logic was that social enterprises, while sharing commonalities with commercial enterprises in willingly taking risks to pursue new value creation, possess a unique characteristic of actively seeking solutions to social issues. Therefore, the leadership style emphasizing an entrepreneurial spirit (i.e., social entrepreneurship) in achieving social objectives is deemed appropriate for social enterprises. However, since Dees and Elias [11] first introduced the concept, social entrepreneurship has not been able to secure clear conceptual independence as a singular leadership style, and there has been an ongoing debate about the theoretical framework for several decades [6,9,12,13].

Other researchers [3,14–17] tried to find the most appropriate leadership style for social enterprises among the main leadership styles for commercial enterprises. However, they faced the problems of not being able to simultaneously compare and analyze various highly regarded leadership styles in commercial enterprises, or of overemphasizing the social aspects of the performance of social enterprises, resulting in a narrow view of their performance.

Therefore, mainstream studies seeking a leadership style suitable for social enterprises have not been able to reach clear conclusions. But the limitations of prior studies do not mean that social enterprises are unnecessary or that leadership is ineffective within them. On the contrary, as the importance and necessity of social enterprises are expected to be increasingly emphasized, efforts to find a leadership style suitable for social enterprises should be continued. In this study, we aim to overcome the shortcomings of previous research by considering various leadership styles simultaneously and balancing performance from both organizational and employee perspectives. Specifically, we will consider four major leadership styles in commercial enterprise to verify which leadership style is appropriate for social enterprise. We use satisfaction variables at the employee level and performance variables at the organizational level as factors for testing the suitability of leadership styles.

## 2. Literature Review and Research Hypothesis

### 2.1. Review of Previous Researches

Social enterprises possess a unique characteristic in valuing both profitability and social benefits, but the importance of leadership to performance improvement is similar to that of commercial enterprises [6,18]. The research on leadership in social enterprises can be broadly categorized into studies identifying leadership styles specialized for social enterprises and studies verifying suitable leadership styles within existing major leadership styles. Prior studies believed in the existence of a unique leadership style specific to social enterprises, focusing on entrepreneurship because they emphasize the importance of a challenging spirit to achieving social goals. Estrin et al. [10] and McMullen [19] insisted that entrepreneurship has profit maximization as its sole goal, while social entrepreneurship pursues the realization of value beyond profit. Raimi et al. [1] also agree that social entrepreneurship can contribute to creating sustainable social impact goals.

However, those claims are still controversial due to the fundamental characteristic of social enterprises, namely, that the values they pursue vary depending on the region or society. Saebi et al. [6], who agreed that social entrepreneurship is important, also emphasized the lack of an integrated conceptual framework for social entrepreneurship. And the studies emphasizing social entrepreneurship claim that social entrepreneurship is one of the best solutions for social enterprises. However, such assertions can be criticized as unrealistic. The effectiveness of leadership can vary depending on how performance is defined; it is more realistic to assume that there can be diverse leadership styles suitable for social enterprises, similar to commercial enterprises.

This research considers social enterprises from various perspectives and attempts to find out which of the effective leadership styles of commercial enterprises is suitable for improving these outcomes.

### 2.2. Outcome of Leadership in Social Enterprise

Over the past 20 years, the question of how to measure the success of social enterprises has become the center of controversy among researchers and practitioners [20]. Many studies on social enterprises examine the result of social enterprises from an organizational perspective, and sustainability is a representative variable. Suriyankietkaew et al. [21] argued that survival should be considered one of the most important goals and outcomes because most social enterprises are small. Survival can be considered the most fundamental factor in that all enterprises, including social enterprises, undergo both establishment and dissolution. However, in this research, survival is inappropriate in that it seeks appropriate leadership for currently surviving social enterprises. Additionally, social enterprises often receive support from government or regulatory agencies for a certain period; thus, there is a problem that survival is possible during that period regardless of the effectiveness of leadership.

Another variable to consider in the success of social enterprises is purpose. Almandoz and Lee [22] insisted that purpose drives the organization of social enterprises and that social enterprise leadership should contribute to achieving the enterprise's purpose. However, because the purpose of an establishment is diverse due to the nature of social enterprises, empirical studies divide the purpose of social enterprises into two categories: economic and social. Bellostas et al. [23] considered economic return and social impact as successful performance in social enterprises in a study targeting sheltered workshops in Spain. Kim [24] considered economic and social performance as markers of the success of social enterprises in a study on the success factors of social enterprises in China. However, considering the results of social enterprises only from the enterprise's perspective may lead to an error of partially assessing the effectiveness of social enterprises. Saebi et al. [6] argued that in order to understand social enterprises more clearly, the employee perspective and the organizational perspective must be considered simultaneously. Chang and Jeong [14] also argued that, considering the characteristics of social enterprises, social enterprises

need leadership that helps them improve performance not only at the organizational level but also at the employee level.

One of the key purposes of social enterprises is to pursue the well-being of employees by providing stable employment [2]; thus, the results related to employees working in social enterprises should also be considered. Iskandar et al. [25] found that employee contributions have a major influence on the performance of social enterprises. Ohana and Meyer [26] regarded efficient management of intention to quit as an important role of leadership in social enterprises, given the high turnover level in these organizations. Given that intention to quit is considered the opposite of satisfaction, their study is regarded as highlighting the importance of employee satisfaction in social enterprises. Cachero-Martínez et al. [27] emphasized that in order for the purchase of a social enterprise to be sustainable, achieving satisfaction through consumption is necessary. They also argued that the experience of satisfaction is important not only for customers but also for employees. Newman et al. [16] found that although servant leadership was positively related to followers' organizational commitment in social enterprises, organizational commitment is a variable that is emphasized more from the perspective of the organization rather than that of the employees, which diverges from the purpose of this study.

Therefore, this study considered job satisfaction and organization satisfaction as perceived performances by employees from an individual level and examined economic performance and social performance from an organizational level.

### 2.3. Relationship with Leadership and Outcome

Recently, some researchers have examined whether the main leadership styles emphasized in commercial enterprises are applicable to social enterprises as well. First, there is an argument that transformational leadership is suitable for social enterprises. Transformational leadership emphasizes transforming individuals into self-motivated beings who recognize the importance of goals, strive to achieve them voluntarily, and are motivated by the leader's intellectual stimulation, encouragement, and charisma, enabling members to reach goals on their own [28]. Bardmili et al. [29] and Gillet et al. [30] considered transformational leadership, which emphasizes change in members to achieve goals, to also be important in social enterprises. Bastari et al. [31] and Naderi et al. [15] also argued that transformational leadership contributes to improving the performance of social enterprises.

Based on their argument, this study considered transformational leadership to be suitable for social enterprises in terms of change and support to achieve goals.

**H1-1.** *Transformational leadership in social enterprises will have a significant relationship with job satisfaction.*

**H1-2.** *Transformational leadership in social enterprises will have a significant relationship with leader satisfaction.*

**H1-3.** *Transformational leadership in social enterprises will have a significant relationship with economic performance.*

**H1-4.** *Transformational leadership in social enterprises will have a significant relationship with social performance.*

Entrepreneurship is highlighted as one of the most-often-considered types of leadership in social enterprises [32]. As social enterprises include the characteristics of commercial enterprises that pursue profitability alongside their unique characteristics of emphasizing social purposes, leaders are required to have a strong spirit of challenge and a sense of responsibility for achieving results [1,6,33,34]. Pinheiro et al. [35] argued that entrepreneurship is important because social enterprises also face pressure to improve performance. Chang and Jeong [14] found that entrepreneurship enhances both the economic and social performance of social enterprises empirically.

Based on their argument, entrepreneurship was considered because challenge is important for balancing economic and social purposes.

**H2-1.** *Entrepreneurship in social enterprises will have a significant relationship with job satisfaction.*

**H2-2.** *Entrepreneurship in social enterprises will have a significant relationship with leader satisfaction.*

**H2-3.** *Entrepreneurship in social enterprises will have a significant relationship with economic performance.*

**H2-4.** *Entrepreneurship in social enterprises will have a significant relationship with social performance.*

To achieve social values, it is crucial to emphasize activities and efforts for the benefit of others over one's own goals. Therefore, it is also necessary to take note of the argument in some studies that servant leadership is a vital role that leaders in social enterprises should perform. Ohana et al. [26] claimed through empirical research that individuals who aspire to join social enterprises are less motivated by monetary factors than those who seek to enter commercial enterprises. In other words, their argument suggests that employees working in social enterprises are more inclined to prioritize social values over personal gains, indicating a greater dedication to others.

Petrovskaya and Mirakyan [36] found that a social enterprise can be successful when all members have a strong sense of service or a calling to realize social value and argued for the leader's dedication to the success of the members as being crucial for performance improvement. Mohamad and Majid [37] also insisted that leaders in social enterprises create conditions for members to excel in their roles. In fact, Newman et al. [16] presented their empirical results that servant leadership enhances organizational effectiveness as perceived by employees in social enterprises.

Based on their argument, servant leadership also focused on the importance of supporting employees from the perspective that employees' success is the success of social enterprises.

**H3-1.** *Servant leadership in social enterprises will have a significant relationship with job satisfaction.*

**H3-2.** *Servant leadership in social enterprises will have a significant relationship with leader satisfaction.*

**H3-3.** *Servant leadership in social enterprises will have a significant relationship with economic performance.*

**H3-4.** *Servant leadership in social enterprises will have a significant relationship with social performance.*

## 3. Research Method and Measure

### 3.1. Participant and Procedure

In studies considering various leadership styles simultaneously, it is necessary to pay attention to the measurement of leadership. Collinson [38] argued that leadership is effective only when a leader has the power to overcome employees' resistance. Alvesson and Spicer [39] supported Collinson's [38] argument, emphasizing the importance of employees perceiving leadership in order words, they suggested that leadership is a product of the interaction between a leader and employees, where the effectiveness of leadership emerges only when employees perceive the leadership of their leader. Therefore, in this study, it was considered that holding the position of leader does not mean that a leader has leadership. So, this study elected to exclude respondents where employees perceived the leadership of their leader to be lower than transactional leadership. In addition, the leadership possessed by a leader can be classified into specific types [14]. But if employees perceive that their leader has leadership, it can be difficult to clearly determine the leader's leadership type because employees can easily make the mistake of perceiving all leadership styles at once [40].

In this study, the survey was conducted twice to solve the problems of prior studies presented above and minimize the bias of the common method. The survey targeted employees of social enterprises who participated in the social enterprise management education program held in Seoul in October 2023. As for the survey method, with the cooperation of the educational institution, the researcher explained the purpose of the study on the first day of the program and then conducted a survey on transactional leadership levels and the four outcome variables for those employees who expressed their willingness

to participate. And on the 15th day, the end date of the program, an additional survey was conducted on the three leadership variables that comprised the independent variables of this study. The first survey had a total of 311 respondents, and the second survey had 302 respondents. In this educational institution, the desk used by a specific person on the first day was to be used until the end of the program. Therefore, the two surveys were integrated easily and used as one. Consequently, the total number of collected samples was 302. Next, in order to analyze employees' perceptions of leaders with leadership, respondents who responded higher than the average based on transactional leadership were presumed to perceive their leader as having leadership. Therefore, 170 surveys in 52 social enterprises that responded to transactional leadership with a score of 3.60 or higher were used as the sample for this study.

### 3.2. Measure

#### 3.2.1. Leadership

This study considered 3 leadership styles as independent variables. Transformational leadership was measured with eight questions according to the concepts and definitions of Bass and Avolio [28]. The Cronbach's alpha for this scale was 0.875. Servant leadership was measured with five questions according to the concepts of Ehrhart [41] and the definitions of Newman et al. [16]. The Cronbach's alpha for this scale was 0.880. And entrepreneurship was measured with five questions according to the concepts and definitions of Helm and Andersson [42]. The Cronbach's alpha for this scale was 0.841. So, the internal consistency reliability of the leadership styles was confirmed (see Table 1).

**Table 1.** Exploratory factor analysis of leadership styles.

| Variable (Cronbach's Alpha) | Item (My Leader) | Factor Loading | | | Source |
|---|---|---|---|---|---|
| Transformational (0.875) | Expresses confidence that goals will be achieved | 0.761 | | | Bass and Avolio (2000) [28] |
| | Emphasizes the importance of a sense of purpose | 0.753 | | | |
| | Talks positively about the future of the enterprise | 0.702 | | | |
| | Helps me see problems from different perspectives | 0.663 | | | |
| | Treats me as a person with different needs, abilities, etc. | 0.654 | | | |
| | Considers the moral/ethical consequences of decisions | 0.633 | | | |
| | Re-examines the main assumptions about the present | 0.601 | | | |
| | Treats me as an individual rather than a member of Org. | 0.553 | | | |
| Servant (0.880) | Has talent that help me heal psychologically | | 0.804 | | Ehrhart (2004) [41] |
| | Helps me with my emotional problems | | 0.779 | | |
| | Sacrifices his/her interests to satisfy my needs | | 0.761 | | |
| | Considers my interests ahead of his/her own | | 0.741 | | |
| | Encourages me to have a sense of community at work | | 0.643 | | |
| Entrepreneurship (0.841) | Has creative solutions to problems | | | 0.802 | Helm and Andersson (2010) [42] |
| | Shows passion for the work | | | 0.775 | |
| | Is willing to take risks | | | 0.723 | |
| | Has a vision for the future of our business | | | 0.589 | |
| | Encourages me to implement more innovative methods | | | 0.488 | |
| | Eigen Value | 8.184 | 1.827 | 1.321 | |
| | % Variance | 45.456 | 10.147 | 7.337 | |
| | Accumulation % | | 55.612 | 62.949 | |

#### 3.2.2. Outcome

This study considered 2 kinds of satisfaction and 2 kinds of performance as the dependent variables. Job satisfaction was measured with 5 questions according to the concepts and definitions of Judge et al. [43]. The Cronbach's alpha for this scale was 0.879. Leader satisfaction was measured with five questions according to the concepts and definitions of Martin et al. [44]. The Cronbach's alpha for this scale was 0.841.

Economic performance and social performance were measured with four questions each according to the concepts and definitions of Bhattarai et al. [45]. Cronbach's alpha for

the two performances was 0.841 and 0.731. So, the internal consistency reliability of the leadership outcomes was also confirmed (see Table 2).

**Table 2.** Exploratory factor analysis of outcomes.

| Variable (Cronbach's Alpha) | Item (I/My Leader/My Enterprise) | Factor Loading | | | | Source |
|---|---|---|---|---|---|---|
| Job Satisfaction (0.879) | Feel quite satisfied with my work | 0.843 | | | | Judge et al. (1998) [43] |
| | Find true joy in my work | 0.815 | | | | |
| | Am almost passionate about my work | 0.806 | | | | |
| | Like my work more than people | 0.727 | | | | |
| | Rarely get bored with my work | 0.673 | | | | |
| Leader Satisfaction (0.841) | Am satisfied working with my leader | | 0.817 | | | Martin et al. (2013) [44] |
| | Like my leader | | 0.806 | | | |
| | Is quite competent in doing his/her work | | 0.767 | | | |
| | Is fair and trustworthy | | 0.709 | | | |
| | Tends to treat his/her subordinates in friendly manner | | 0.529 | | | |
| Economic Performance (0.841) | Has achieved its profit goals | | | 0.885 | | Bhattarai et al. (2019) [45] |
| | Has improved the level of management independence | | | 0.858 | | |
| | Has continued to increase sales | | | 0.829 | | |
| | Has increased customer satisfaction with our products/services | | | 0.668 | | |
| Social Performance (0.731) | Invests enterprise's profits in public interest projects/programs | | | | 0.770 | |
| | Contributes to the stabilization of society | | | | 0.710 | |
| | Focuses on preserving the environment and creating jobs | | | | 0.703 | |
| | Eigen Value | 6.305 | 2.684 | 1.563 | 1.124 | |
| | % Variance | 37.086 | 15.788 | 9.196 | 6.611 | |
| | Accumulation % | | 52.875 | 62.071 | 68.681 | |

For all measures, participants rated items using a 5-point Likert scale where 1 = 'strongly disagree' and 5 = 'strongly agree'.

### 3.3. Data Analysis

Prior to testing hypotheses, exploratory factor analysis (EFA) and principal component factor analysis were conducted using SPSS 29 to examine the factor structure of the measures. The questions were grouped in terms of significance by the variable to be measured, and the seven values also exceeded 1 (transformational leadership = 8.184; servant leadership = 1.827; entrepreneurship = 1.321; job satisfaction = 6.305; leader satisfaction = 2.684; economic performance = 1.563; social performance = 1.124). All questions, except one about social performance ('the social enterprise I work for is recognized by the local community'), were grouped into one variable. Tables 1 and 2 report the results of exploratory factor analysis for the measured variables.

## 4. Result

### 4.1. Correlation

Table 3 shows the means, standard deviations, and Spearman correlations of the variables. The means and standard deviations of the variables ranged from 3.346 to 3.813 and 0.615 to 0.882, respectively. The correlation coefficients revealed that all variables are distinct, and the highest correlation exists between leader satisfaction and servant leadership ($r = 0.730$; $p < 0.01$). All relationships between variables were confirmed to be as expected, with positive relationships.

**Table 3.** Correlation matrix and descriptive statistics.

| | 1 | 2 | 3 | 4 | 5 | 6 | 7 |
|---|---|---|---|---|---|---|---|
| Economic Performance (1) | 1 | | | | | | |
| Social Performance (2) | 0.247 ** | 1 | | | | | |
| Job Satisfaction (3) | 0.241 ** | 0.469 ** | 1 | | | | |
| Leader Satisfaction (4) | 0.087 | 0.544 ** | 0.567 ** | 1 | | | |

**Table 3.** *Cont.*

| | 1 | 2 | 3 | 4 | 5 | 6 | 7 |
|---|---|---|---|---|---|---|---|
| Transformational Leadership (5) | 0.203 ** | 0.515 ** | 0.572 ** | 0.679 ** | 1 | | |
| Entrepreneurship (6) | 0.201 ** | 0.467 * | 0.554 ** | 0.638 ** | 0.666 ** | 1 | |
| Servant Leadership (7) | 0.043 ** | 0.541 ** | 0.471 ** | 0.730 ** | 0.601 ** | 0.602 ** | 1 |
| Mean | 3.541 | 3.406 | 3.665 | 3.667 | 3.813 | 3.719 | 3.346 |
| S.D. | 0.765 | 0.797 | 0.782 | 0.767 | 0.616 | 0.719 | 0.882 |

Note: ** $p < 0.01$; * $p < 0.05$ (two-tailed).

### 4.2. Regression

Regression analysis was conducted to examine the effect of leadership styles in social enterprises as presented in Table 4. Models 1 and 2 in Table 4 present the analysis results of the relationship between leadership and satisfaction. Job satisfaction showed a significant relationship with transformational leadership ($\beta = 0.364$; $p < 0.001$) and entrepreneurship ($\beta = 0.274$; $p < 0.01$). Among the hypotheses regarding job satisfaction, 1-1 and 2-1 were accepted. Leader satisfaction showed a significant relationship with transformational leadership ($\beta = 0.336$; $p < 0.001$), entrepreneurship ($\beta = 0.117$; $p < 0.05$), and servant leadership ($\beta = 0.414$; $p < 0.001$). Among the hypotheses regarding leader satisfaction, 1-2, 2-2, and 3-2 were accepted.

**Table 4.** Leadership and outcome.

| Dependent Variable: | Job Satisfaction | | Leader Satisfaction | | Economic Performance | | Social Performance | |
|---|---|---|---|---|---|---|---|---|
| | Coefficient (β) | *t*-Value | Coefficient (β) | *t*-Value | Coefficient (β) | *t*-Value | Coefficient (β) | *t*-Value |
| Gender | 0.057 | 0.882 | −0.088 * | −1.714 | −0.163 * | −1.958 | −0.148 * | −2.155 |
| Age | 0.195 ** | 3.006 | 0.067 | 1.291 | 0.045 | 0.540 | −0.067 | −0.971 |
| Education | 0.022 | 0.354 | 0.015 | 0.307 | −0.023 | −0.291 | 0.058 | 0.885 |
| Position | −0.198 ** | −2.740 | −0.060 | −1.048 | −0.041 | −0.441 | −0.181 * | −2.346 |
| Tenure | −0.028 | −0.421 | −0.002 | −0.038 | 0.126 | 1.476 | −0.013 | −0.180 |
| N of Employee | 0.052 | 0.738 | −0.109 * | −1.924 | 0.055 | 0.601 | −0.047 | −0.616 |
| Enterprise Type | −0.130 * | −1.854 | 0.022 | 0.396 | −0.103 | −1.139 | 0.058 | 0.771 |
| Industry | −0.030 | −0.533 | −0.042 | −0.879 | −0.122 | −1.581 | 0.072 | 1.130 |
| Transformational Leadership | 0.364 *** | 4.191 | 0.336 *** | 4.847 | 0.266 ** | 2.370 | 0.319 *** | 3.433 |
| Entrepreneurship | 0.274 ** | 3.243 | 0.117 * | 1.737 | 0.166 | 1.524 | 0.049 | 0.545 |
| Servant Leadership | 0.023 | 0.269 | 0.414 *** | 6.150 | −0.161 | −1.480 | 0.224 ** | 2.481 |
| *Adj.R²* | 0.449 | | 0.650 | | 0.085 | | 0.372 | |
| F | 13.539 | | 29.484 | | 2.436 | | 10.109 | |

Note: *** $p < 0.001$; ** $p < 0.01$; * $p < 0.05$ (one-tailed).

Models 3 and 4 in Table 4 present the analysis results of the relationship between leadership and performance. Economic performance only showed a significant relationship with transformational leadership ($\beta = 0.266$; $p < 0.01$). Social performance showed a significant relationship with transformational leadership ($\beta = 0.319$; $p < 0.001$) and servant leadership ($\beta = 0.224$; $p < 0.01$). Among the performance hypotheses, 1-3 and 1-4 (regarding transformational leadership) were accepted, and 3-4 (regarding servant leadership) was also accepted.

### 4.3. Additional Analysis

In order to verify the mechanism or process of the relationship between leadership and outcomes more clearly, this study conducted additional analysis to examine the mediating effect of two kinds of satisfaction. The mediating effect was confirmed through Tables 4 and 5, applying the method of Baron and Kenny [46]. As presented in Tables 4 and 5, job satisfaction partially mediated the effect of transformational leadership on economic and social performance.

**Table 5.** Additional analysis: mediating effect of satisfaction.

| Dependent Variable: | Economic Performance | | | | Social Performance | | | |
|---|---|---|---|---|---|---|---|---|
| | Coefficient (β) | *t*-Value | Coefficient (β) | *t*-Value | Coefficient (β) | *t*-Value | Coefficient (β) | *t*-Value |
| Gender | −0.174 * | −2.105 | −0.172 * | −2.057 | −0.159 * | −2.340 | −0.137 * | −1.978 |
| Age | 0.007 | 0.080 | 0.053 | 0.627 | −0.105 | −1.491 | −0.076 | −1.088 |
| Education | −0.027 | −0.348 | −0.021 | −0.270 | 0.053 | 0.832 | 0.056 | 0.857 |
| Position | −0.002 | −0.023 | −0.048 | −0.512 | −0.143 * | −1.834 | −0.173 * | −2.242 |
| Tenure | 0.131 | 1.552 | 0.125 | 1.472 | −0.007 | −0.105 | −0.012 | −0.177 |
| N of Employee | 0.044 | 0.492 | 0.042 | 0.461 | −0.057 | −0.758 | −0.033 | −0.431 |
| Enterprise Type | −0.077 | −0.855 | −0.100 | −1.110 | 0.083 | 1.107 | 0.055 | 0.734 |
| Industry | −0.116 | −1.512 | −0.127 | −1.638 | 0.078 | 1.241 | 0.077 | 1.211 |
| Transformational Leadership | 0.194 * | 1.656 | 0.303 ** | 2.525 | 0.249 ** | 2.575 | 0.276 ** | 2.781 |
| Entrepreneurship | 0.112 | 1.005 | 0.179 | 1.629 | −0.004 | −0.038 | 0.034 | 0.379 |
| Servant Leadership | −0.165 | −1.533 | −0.114 | −0.942 | 0.219 ** | 2.464 | 0.171 * | 1.711 |
| Job Satisfaction | 0.197 * | 1.934 | | | 0.192 * | 2.295 | | |
| Leader Satisfaction | | | −0.113 | −0.878 | | | 0.126 | 1.183 |
| *Adj.R*$^2$ | 0.165 | | 0.084 | | 0.389 | | 0.374 | |
| F | 2.583 | | 2.294 | | 9.956 | | 9.407 | |

Note: ** $p < 0.01$; * $p < 0.05$ (one-tailed).

And the results of the empirical analysis that examined the effect of three leadership styles and leader satisfaction on economic and social performance were not sufficient to satisfy the verification conditions of Baron and Kenny [46]. So, this study did not find a mediating effect of leader satisfaction.

## 5. Discussion

The key finding of these results is that transformational leadership is important for achieving various outcome improvements not only in commercial enterprises but also in social enterprises. In this study, transformational leadership showed significant relationships with all four variables (job satisfaction, leader satisfaction, economic performance, social performance) of leadership outcome considered at both the individual and corporate levels. This finding is similar to the position of prior studies that concern the importance of enhancing outcomes [15,29–31]. Specially, Suriyankietkaew et al. [21] identified four critical factors of sustainable leadership in social enterprises through the qualitative method, of which three were similar to the characteristics of transformational leadership, excluding ethical competencies. Their arguments, and this study, enable future researchers to gain insight into the importance of transformational leadership in improving outcomes, regardless of whether enterprises pursue profit or not. And all leadership styles considered in this study influenced employees' leader satisfaction. These results indicate that the major leadership styles of commercial enterprises also operate effectively in social enterprises.

Another insightful finding of this study is that entrepreneurship can influence individual-level outcomes but not corporate-level outcomes, and servant leadership can help social enterprises to achieve social purposes. These imply that the appropriate leadership in social enterprises may vary depending on what the outcome is. This is similar to the position of studies that emphasize the challenge of new approaches [14,35].

This study also provides some interesting results that differ from previous research. Saebi et al. [6] argued that the tension between economic and social missions is necessary for social enterprises to succeed, and that the key to achieving the goal is entrepreneurship. Chang and Jeong [14] also empirically confirmed that entrepreneurship has a positive impact on improving various performances in social enterprises. But in this study, entrepreneurship did not show a significant relationship with economic and social performances. The reason for this could first be found in the methodology of this study. This study attempted to control the impact of leadership on outcomes regardless of the leadership type; therefore, it only sampled respondents who perceived transaction leadership

to be higher than the average value. And the emphasis on challenge, which is generally assumed to have a negative or no relationship with performance, may be the reason why the relationships between entrepreneurship and performance were insignificant.

Newman et al. [16] emphasized the importance of servant leadership and argued that when commitment to employees is possible, dedication to society is also possible. Petrovskaya and Mirakyan [36] pointed out that specific leadership in social enterprises is important for supporting employee success rather than driving employee behavior. But in this study, servant leadership did not impact job satisfaction and economic performance. It could be assumed that these results are due to the unique characteristics of servant leadership, which focuses on humans. In Newman et al. [16]'s study, servant leadership improved performance related to interpersonal relationships but was not significant for work-related performance. The study by Petrovskaya and Mirakyan [36] focused on revealing the differences between commercial and social enterprises but did not clarify their relationship with performance.

## 6. Conclusions

In proportion to the increasing academic interest in social enterprises, research on leadership in social enterprises has also been conducted extensively [16,45]. Research on leadership in social enterprises initially focused on discovering new leadership styles, such as social entrepreneurship. However, it has not yet established a clear and dominant framework related to that [6]. Recently, research has been conducted to identify which types of leadership are more effective for social enterprises [17,29,32]. Naderi et al. [15] suggested transformational leadership, Newman et al. [16] and Petrovskaya and Mirakyan [36] emphasized servant leadership, and Pasricha et al. [17] highlighted ethical leadership as the suitable leadership styles for social enterprises.

The research gap that this study fills is that prior studies have either focused only on the specific leadership styles of commercial enterprises or failed to consider various outcomes for social enterprises. The present study aimed to confirm empirically whether the three major leadership styles of commercial enterprises contribute to improving the performance of social enterprises at individual and firm levels. In particular, this study used only samples in which leadership was consistently effective in order to figure out the true impact of each leadership style on outcomes in social enterprises. Analyzing data from 170 surveys in 52 social enterprises, the present study found that transformational leadership was positively related to all outcomes in social enterprises, entrepreneurship was positively related to satisfaction as an individual-level outcome, and servant leadership was positively related to interpersonal outcomes as a firm-level outcome. These results demonstrated that effective leadership styles in commercial enterprises also work effectively in social enterprises. Specially, the general leadership style for improving outcomes in social enterprises is transformational leadership. And this study showed that appropriate leadership can be either entrepreneurship or servant leadership depending on what objectives of social enterprises one aims to enhanced.

Although this study contributes to the direction of future research on suitable leadership styles in social enterprises by considering the relationship between various leadership styles and outcome, it has some limitations. The explanatory value of the effects of three leadership styles on economic performance was observed to be low. This result is estimated to have arisen from issues with the sample, such as using only respondents that perceived transactional leadership to be above average in statistical analysis. As a result, it can be seen that the impact of the three leadership styles on economic performance is minimized.

Another reason why improvement via leadership styles in economic performance was not well observed in this study could be found in the role of social enterprises in South Korea. In South Korea, the level of economic independence of social enterprises is so low that many social enterprises go out of business when government support stops. As a result, employees of social enterprises tend not to evaluate performance highly. To overcome these limitations, it seems necessary to conduct research targeting employees

of social enterprises in other countries. In future research, it may be necessary to consider measuring objective performance rather than subjective performance as one way to solve these issues.

Another limitation of this study is its failure to provide clearer evidence regarding what constitutes the primary leadership styles in commercial enterprises. If this study had provided reasons as to why the three leadership styles considered here, rather than recently emphasized leadership styles such as authentic leadership, are more important, or provided evidence supporting their status as the main leadership styles, the implications of this study could have been better elucidated.

The failure to elucidate the impact of entrepreneurship on performance is considered a significant limitation of this study. In regression analysis with only entrepreneurship as input, a significant positive relationship was observed. However, in regression analysis with all three leadership styles included, the relationship between entrepreneurship and performance was not significant. One possible explanation is that the impact of other leadership styles on performance is significant, which may have offset the effect of entrepreneurship on performance. However, further research is needed to verify this.

Additionally, there is a need to study the relationship between leadership and performance by attempting a new approach to understanding leadership effectiveness, such as leadership learning (Guterresa et al. [47]) or networking (Liu et al. [48]). Finally, a limitation of this study is the lack of generalizability of the research results due to characteristics of self-assessment surveys such as common method bias and targeted sampling.

**Funding:** This research was funded by Kyungnam University Foundation Grant, 2021.

**Institutional Review Board Statement:** Ethical review and approval of this study were waived due to only use questionnaires that had been approved on ethical grounds in previous research.

**Informed Consent Statement:** Informed consent was obtained from all subjects involved in the study.

**Data Availability Statement:** Data are available upon reasonable request.

**Conflicts of Interest:** The author declares no conflicts of interest.

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
