# Peer review of "The Relationship between Leadership and Performance in Enhancing the Sustainability of Social Enterprises"

_sustainability, doi:10.3390/su16083218_

Round 1

Reviewer 1 Report

Comments and Suggestions for Authors

This work addresses a topic that has increasingly gained relevance, namely social enterprises, and particularly the issue of leadership in this type of companies, adding some knowledge on this topic.

The work is, therefore, relevant for the field. It presents an adequate structure, although, some of the sections could be improved, as described below.

 The abstract presents the aim of the study, the methodology and the main results achieved. However, the methodology used is not clearly presented, and should be improved. The abstract begins with a “strong” statement that should be justified, at least supported with a citation (“Despite significant interest in social enterprises over recent decades, research on finding suitable leadership for social enterprises has been inadequate”).

The literature review presents the main concepts, but the gap that the authors want to fill is not evident, nor why authors say that research on finding suitable leadership for social enterprises has been inadequate. Although many citations are very relevant and recent, there are some not so recent; some are cited as if they were current, but in fact, given the topic, their relevance is questioned (examples: 3, 6, 7, 8, 9, 10, 13, 28).

The research method seems adequate and appropriate to the study. However, the way in which the survey was applied and to whom is a little confusing, and a rewrite of section 3.1 is recommended.

Authors should also better justify the criteria to exclude leaders without leadership. This exclusion seems to bring limitations to the work, as authors present in the conclusion, therefore it is important to clarify it.

The design of the questionnaire, to measure Transformational leadership (section 3.2.1) is based on work from 2000; one wonders whether it will be the most appropriate; It would be important in this section to indicate what type of scale was used.

The need for the characteristics described in table 1 is also not evident for this study, as they are not considered in the discussion.

Regarding the keywords, maybe delete satisfaction and performance as those words are too general.

Comments on the Quality of English Language

The article has some minor errors (lines 35, 46) and some slightly confusing sentences.

Author Response

The main argument of this study is that efforts to identify suitable leadership styles for social enterprises should first confirm whether the major leadership style recognized in commercial enterprises effectively operates in social enterprises, rather than seeking new leadership styles.

It seems that this research purpose was not properly emphasized during the process of writing the paper, causing confusion. The paper was revised to reflect your comments as much as possible. The revised points can be summarized as follows:

Comments # The abstract begins with a “strong” statement that should be justified, at least supported with a citation (“Despite significant interest in social enterprises over recent decades, research on finding suitable leadership for social enterprises has been inadequate”).

Line 7-8: Edit wording

Despite significant interest in social enterprises over recent decades, research on finding suitable leadership for social enterprises has been inadequate.

  • Despite considerable interest in social enterprises over the past few decades, no consistent conclusion has been reached about what leadership style is appropriate for social enterprises.

Line 16-18: Edit wording

In this study, to overcome another limitation of prior studies that narrowly defined the outcomes of leadership, job satisfaction and leader satisfaction were considered at the individual level, while economic performance and social performance were considered at the firm level.

  • In this study, in order to broadly understand the performance of leadership, job satisfaction and leader satisfaction were considered at the individual level, and economic performance and social performance were considered at the firm level.

Comments # The literature review presents the main concepts, but the gap that the authors want to fill is not evident, nor why authors say that research on finding suitable leadership for social enterprises has been inadequate.

Line 8 – 10: Insert sentence

The present study aimed to find suitable leadership for social enterprises within the primary leadership styles recognized in commercial enterprises.

Comments # the methodology used is not clearly presented, and should be improved.

Line 17 – 18: Edit wording

To more clearly investigate the claims of this study, a survey on transactional leadership was conducted, and 15 days later,

  • To support the argument of this study, it aimed to survey employees who perceive a leader as possessing leadership qualities in social enterprises. For this purpose, a survey on transaction leadership was conducted among employees who participated in three-week training section related social enterprises. And the last day of training,

Comments # There are some not so recent; some are cited as if they were current, but in fact, given the topic, their relevance is questioned (examples: 3, 6, 7, 8, 9, 10, 13, 28).

  • I also agree with your comment. Some of the research cited in this study may be older, but they are mentioned in the text due to their traditional emphasis on leadership. In future research, the focus will be on citing more recent studies.

Comments # The way in which the survey was applied and to whom is a little confusing, and a rewrite of section 3.1 is recommended. Authors should also better justify the criteria to exclude leaders without leadership. This exclusion seems to bring limitations to the work, as authors present in the conclusion, therefore it is important to clarify it.

  • I completely agree with your comment. I particularly agree that I need to write the wording of Section 3.1 more clearly and clarify the reasons for removing leaders without leadership entirely.
  • So, I revised section 3.1 by adding studies suggesting that employees must perceive leadership in order for leadership to have an effect.

3.1 Section

Line 227: Insert sentence

  • Collinson (2005) argued that leadership is effective only when a leader has the power to overcome employees’ resistance [38]. Alvesson and Spicer (2012) supported Collinson’s (2005) argument, emphasizing the importance of employees perceiving leadership [39]. In order words, they suggest that leadership is a product of interaction between a leader and employees, where the effectiveness of leadership emerges only when employees perceive the leadership of their leader. Therefore, in this study, it was considered that holding a position of the leader does not mean that a leader has leadership. So, this study determined to exclude respondents where employees perceived the leadership of their leader to be lower than transactional leadership.

Line 228-229: Delete sentence

First, just because the job position is leader does not mean that the person has leadership, therefore, responses regarding leaders without leadership should be excluded.

Comments # The need for the characteristics described in table 1 is also not evident for this study, as they are not considered in the discussion.

  • I completely agree with your comment. However, when in another study submission to this journal, reviewers requested detailed information about the sample, so I have included table 1 in this study as well.
  • Table 1 was deleted, and the table numbers were also revised.

Comments # The article has some minor errors (lines 35, 46) and some slightly confusing sentences.

Line 35~36: Edit wording

However, as social issues of interest vary by region, social enterprises take on a wide range of forms by reflecting the characteristics of the specific locality

  • However, since social concerns vary by region, social enterprises can pursue various forms or different objectives reflecting the characteristics of the region.

Line 45-46: Edit wording

These individuals asserted that social enterprises do not emphasize presenting a universally interpreted, clear goa, or orientation that is the same for everyone, regardless of the observer.

  • These researchers argued that social enterprises do not universally interpret or provide clear goals or direction to everyone regardless of observers.

Comments # Regarding the keywords, maybe delete satisfaction and performance as those words are too general.

  • As your comment, satisfaction and performance were removed from the keywords.

Line 34: Delete keywords

  • transformational leadership; entrepreneurship; servant leadership; social enterprises

(, satisfaction; performance)

Sincerely.

Reviewer 2 Report

Comments and Suggestions for Authors

Congratulations for your article. However, I think the paper needs some improvements, especially in the way the results and discussion are presented:

·      Introduction: Comprehensive, with recent and relevant quotations.

·      Literature review and hypotheses: Although I note an adequate review, I do not entirely agree with introducing hypotheses where you do. Sometimes it seems to create the confusion of being hypotheses that arise from the very studies you cite throughout the paragraphs. Hypotheses would be introduced in a convenient section within the methodology and, furthermore, all of them should be addressed in the results, discussion, or conclusions. After reading the text I do not see a direct relationship between the content from the results and discussion section and the hypotheses put forward. Furthermore, I would synthesize them, given the subsequent analysis, they seem too many hypotheses.

·      Methodology: Use of an experimental design appropriate to the stated research objectives, with clear, obvious, and detailed explanations. However:

o   Are there any limitations? I would introduce a short paragraph on the limitations of the study. I would urge to identify the weaknesses of the study as reader also wishes to know what difficulties and challenges for you to conduct such study. 

o   Are there any ethical considerations?

·      Findings/Results: Clear and detailed. However no clear relation with hypothesis.

·      Discussion: Complete and referenced with previous literature. However:

o   I would introduce an introductory paragraph briefly summarizing the aim of the research and the methodologies applied.

o   I do not find the hypotheses answered or addressed in the discussion. Whether in the results or in this section, it would be very necessary to organize them around the hypotheses raised.

·      Conclusion: Fine with them.

·      Are there policy or practical implications of this study? Another section addressing this question would be very appropriate.

·      Tables and figures: They are appropriate and very clear. They are easy to interpret, and they support the arguments. However, I would add the source (are they own elaborated, right?).

Finally, congratulations for the article that, with further improvements, may be considered for publication. I hope my comments will be helpful. Especially, pay attention to the hypotheses and to the writing of results and discussion.

Sincerely.

Author Response

The main argument of this study is that efforts to identify suitable leadership styles for social enterprises should first confirm whether the major leadership style recognized in commercial enterprises effectively operates in social enterprises, rather than seeking new leadership styles.

It seems that this research purpose was not properly emphasized during the process of writing the paper, causing confusion. The paper was revised to reflect your comments as much as possible. The revised points can be summarized as follows:

Comments # I do not entirely agree with introducing hypotheses where you do. Sometimes it seems to create the confusion of being hypotheses that arise from the very studies you cite throughout the paragraphs.

Comments # After reading the text I do not see a direct relationship between the content from the results and discussion section and the hypotheses put forward. Furthermore, I would synthesize them, given the subsequent analysis, they seem too many hypotheses.

  • Thanks for your comments. I attempted to clearly present the arguments of this study, but I believe improvements are necessary as it has caused confusion. However, another reviewer presented more diverse opinions on the relationship between variables through a review of previous research. Therefore, Therefore, before suggesting hypotheses, I included a brief summary statement outlining the arguments of this study.
  • So, I ask for your understanding that the attempt to demonstrate various perspectives on the arguments of this study has resulted in such confusion.

Line 171: Insert sentence

  • Based on their argument, this study considered transformational leadership to be suitable for social enterprises in terms of change and support to achieve goals.

Line 188: Insert sentence

  • Based on their argument, the entrepreneurship was considered in that challenge is important for balancing economic and social purposes.

Line 213: Insert sentence

  • Based on their argument, servant leadership also focused on the importance of support for employees from the perspective that employees’ success is the success of social enterprises.

Line 171-172: Delete sentence

  • Hypothesis 1. The transformational leadership of a social enterprise leader will have a significant relationship with outcomes of social enterprises.

Line 189-190: Delete sentence

  • Hypothesis 2. The entrepreneurship of a social enterprise leader will have a significant relationship with outcomes of social enterprises.

Line 214-215: Delete sentence

  • Hypothesis 3. The servant leadership of a social enterprise leader will have a significant relationship with outcomes of social enterprises.

Comments # Especially, pay attention to the hypotheses and to the writing of results and discussion.

  • I completely agree with your comment. As important as the objectives and results of the study are, the limitations of the research are equally crucial, but it was not mentioned much.

Line 422: Insert sentences

  • Another reason why the improvement by leaderships in economic performance was not well observed in this study could also be found in the role of social enterprises in South Korea. In South Korea, the level of economic independence of social enterprises is so low that many social enterprises go out of business when government support stops. As a result, employees of social enterprises tend to not evaluate performance highly. To overcome these limitations, it seems necessary to conduct research targeting employees of social enterprises in other countries. In future research, it may be necessary to consider measuring objective performance rather than subjective performance as one way to solve these issues.
  • Another limitation of this study is the failure to provide clearer evidence regarding what constitutes the primary leadership in commercial enterprises. If this study had provided reasons why the three leadership styles considered here, rather than recently emphasized leadership such as authentic leadership, are more important or the evidence claiming them as the main leadership, the implications of this study could have been better elucidated.
  • The failure to elucidate the impact of entrepreneurship on performance is considered a significant limitation of this study. In regression analysis with only entrepreneurship input, a significant positive relationship was observed. However, in regression analysis with all three leadership styles included, the relationship between entrepreneurship and performance was not significant. One possible explanation is that the impact of other leadership styles on performance is significant, which may have offset the effect of entrepreneurship on performance. However, further research is needed to verify this.
  • Additionally, there is a need to study the relationship between leadership and performance by attempting a new approach to leadership effectiveness, such as leadership learning [48] or networking [49].

Comments # However, I would add the source (are they own elaborated, right?).

  • The survey items of this study were adopted from those used in relevant research and noted in the '3.2 section.' However, as per your suggestion, since it seems difficult to verify the survey source, the sources of the surveys are added in tables 1 and 2.
  • Thanks for your comments.

Sincerely,

Reviewer 3 Report

Comments and Suggestions for Authors

Thanks the opportunity to review this interesting study of "Does the Leadership of Commercial Enterprises also Contribute to the Improvement of Social Enterprises' Outcomes?".

As I see it delves into the intricate relationship between leadership styles in enterprises and their potential influence on the performance of social enterprises. By scrutinizing various leadership approaches, the research seeks to uncover the nuanced ways in which these styles may contribute to or impede the advancement of social enterprises. However, to enhance the clarity of the paper's positioning, there is a need for a more explicit delineation of the study's objectives and its significance within the broader context of leadership studies, such as leadership styles. As an example you are testing styles of e.g. Transformational Leadership vs. Entrepreneurial Leadership vs. Servant Leadership in social enterprices. In addition in the conclusion you need to give more clear managerial and theoretical implication of these interesting findings.

However there are some methodological  concerns. It is imperative to incorporate explicit meta-communication within the text, elucidating the significance of this chosen sample in relation to the broader population. Clarifying the rationale for the sample's relevance will enhance the transparency and validity of the study, fostering a more comprehensive understanding of its findings in the context of the larger population under investigation.

Finally, it is essential to address the numerous minor errors present in the text, as they detract from its overall polish. A thorough and meticulous revision is necessary to rectify these errors, ultimately elevating the professionalism and readability of the paper. This concerted effort will ensure that the focus remains squarely on the substantive aspects of the research, rather than being inadvertently overshadowed by the mechanics of the writing.

In summary, these recommendations are designed to enhance the study's theoretical and managerial clarity, bolster methodological robustness, and refine overall presentation. Implementing these suggestions is anticipated to maximize the study's contribution to the field of leadership studies and significantly augment its impact on our understanding of social enterprise outcomes.

I look forward to the continued refinement of this potential valuable contribution.

Comments on the Quality of English Language

Needs to improved with minor errors present in the text. But overall good written english.

Author Response

The main argument of this study is that efforts to identify suitable leadership styles for social enterprises should first confirm whether the major leadership style recognized in commercial enterprises effectively operates in social enterprises, rather than seeking new leadership styles.

It seems that this research purpose was not properly emphasized during the process of writing the paper, causing confusion. The paper was revised to reflect your comments as much as possible. The revised points can be summarized as follows:

Comments # However, to enhance the clarity of the paper's positioning, there is a need for a more explicit delineation of the study's objectives and its significance within the broader context of leadership studies, such as leadership styles.

  • I completely agree with your comment. However, another reviewer requested that the number of references be reduced, saying that the researcher's claims seemed rather confusing due to the large number of prior studies. So, I added the content of broader research on leadership styles to the conclusion section rather than the hypothesis section.
  • Thanks for your comments.

Comments # Clarifying the rationale for the sample's relevance will enhance the transparency and validity of the study, fostering a more comprehensive understanding of its findings in the context of the larger population under investigation.

  • I completely agree with your comment. So, I revised section 3.1 by adding studies suggesting that employees must perceive leadership in order for leadership to have an effect.

3.1 Section

Line 227: Insert sentences

  • Collinson (2005) argued that leadership is effective only when a leader has the power to overcome employees’ resistance [38]. Alvesson and Spicer (2012) supported Collinson’s (2005) argument, emphasizing the importance of employees perceiving leadership [39]. In order words, they suggest that leadership is a product of interaction between a leader and employees, where the effectiveness of leadership emerges only when employees perceive the leadership of their leader. Therefore, in this study, it was considered that holding a position of the leader does not mean that a leader has leadership. So, this study determined to exclude respondents where employees perceived the leadership of their leader to be lower than transactional leadership.

Line 228-229: Delete sentence

  • First, just because the job position is leader does not mean that the person has leadership, therefore, responses regarding leaders without leadership should be excluded.

Comments # Finally, it is essential to address the numerous minor errors present in the text, as they detract from its overall polish.

  • Thanks for your comments. I have revised typos and awkward expressions along with comments by the anonymous reviewer. The revised sentences or references are highlighted in red.

Sincerely.

Reviewer 4 Report

Comments and Suggestions for Authors

Dear Sir,

I think that this paper have a Percent match: 24%, which cannot be accepted for publication in this Journal.

I think that this paper need major revisions.

All the best 

Author Response

The main argument of this study is that efforts to identify suitable leadership styles for social enterprises should first confirm whether the major leadership style recognized in commercial enterprises effectively operates in social enterprises, rather than seeking new leadership styles.

It seems that this research purpose was not properly emphasized during the process of writing the paper, causing confusion. The paper was revised to reflect your comments as much as possible.

But, first, I would like to thank the reviewers for their interest in reviewing this study. The issue of plagiarism is a very serious issue in research ethics, and a 24% plagiarism rate is truly serious.

I reviewed various previous studies and wrote the sentences myself, albeit poorly, in English, so the 24% plagiarism rate was very shocking.

The plagiarism check program I have access to showed a plagiarism rate of less than 2%, so if you don't mind being rude, I would be grateful if you could send me the test results.

I am sorry for causing concern to the reviewer with this issue.

Sincerely.

Reviewer 5 Report

Comments and Suggestions for Authors

The author continues his research published in the journal in 2021 (Does Leadership Matter in the Performance of Social Enterprises in South Korea?, https://www.mdpi.com/2071-1050/13/20/11109). The studies have some stylistic similarities. The anti-plagiarism check did not reveal any significant comments. At the same time, the authors need to check again.

The relevance of the topic of the article is due to the growing interest in social enterprises in recent decades and the insufficient development of leadership issues for social enterprises. The study aims to examine the impact of three main types of leadership on satisfaction and performance of social enterprises.

In a review of the scientific literature, the author rightly highlights the insufficiency of previous research related to the definition of effective leadership for social enterprises. But at the same time, there are no publications from recent years (in references). Particular attention is paid to the analysis of transformational leadership, entrepreneurship and servant leadership in the context of their potential impact on social enterprises. 

The study demonstrates that leadership adopted in commercial enterprises can be effectively applied in social enterprises. Transformational leadership plays a particularly important role. The results obtained can be used to develop management strategies in social enterprises.

However, the study has a number of areas for improvement, including poor explainability of the impact of leadership on economic performance. Also, the relationship between entrepreneurial leadership and organizational performance has been found to be mixed. At these points it is necessary to strengthen the analysis and provide comments in the article.

We see that the survey was conducted at social enterprises in South Korea. Can the findings be extended to other countries? What additional assumptions might there be in this?

The leadership classification used, enshrined in the hypotheses, is not the only one. There are many approaches to this issue. Therefore, the authors need additional arguments in favor of choosing these three types of leadership for analysis.

The article would have benefited from a more detailed description of the research results in terms of their contribution to understanding the mechanisms of interaction between leadership and organizational performance in social enterprises. It is advisable to provide more detailed practical recommendations for managers of social enterprises.

The final recommendation addresses the sustainability context. The article was submitted to the journal Sustainability, and therefore should cover sustainability issues. However, the context of sustainability is not disclosed in the text of the article. The authors are recommended to supplement the text of the Introduction and other sections of the article with a connection to sustainability. This will also require some changes to the title of the article.

For this type of research, it is good practice to disclose the aggregated database in an anonymous form and post it in a repository so that readers can follow the authors' path (Data Availability Statement). This also increases the reliability of the findings.

The article makes a contribution to research on the relationship between leadership and performance in social enterprises. Despite some limitations, the article provides valuable directions for further research in this area and may contribute to the development of effective management strategies in social enterprises.

Author Response

The main argument of this study is that efforts to identify suitable leadership styles for social enterprises should first confirm whether the major leadership style recognized in commercial enterprises effectively operates in social enterprises, rather than seeking new leadership styles.

It seems that this research purpose was not properly emphasized during the process of writing the paper, causing confusion. The paper was revised to reflect your comments as much as possible. The revised points can be summarized as follows:

Comments # The study has a number of areas for improvement, including poor explainability of the impact of leadership on economic performance.

Comments # The relationship between entrepreneurial leadership and organizational performance has been found to be mixed. At these points it is necessary to strengthen the analysis and provide comments in the article.

Comments # Can the findings be extended to other countries? What additional assumptions might there be in this? What additional assumptions might there be in this?

  • Thanks for your accurate comments as an expert. I completely agree with your comments. I also thought that entrepreneurship would have a significant positive relationship with organizational performance, but the results were not significant.
  • One of the major limitations of this study is that the relationship between leadership and performance was studied only in South Korean social enterprises. To overcome these limitations, I personally plan to start research in Los Angeles in August of this year. I intend to study how leadership achieves certain outcomes in social enterprises with diverse racial and national backgrounds in LA.
  • The limitations of the sample and the need for expansion were mentioned in the discussion.

Line 435: Insert sentences

  • Another reason why the improvement by leaderships in economic performance was not well observed in this study could also be found in the role of social enterprises in South Korea. In South Korea, the level of economic independence of social enterprises is so low that many social enterprises go out of business when government support stops. As a result, employees of social enterprises tend to not evaluate performance highly. To overcome these limitations, it seems necessary to conduct research targeting employees of social enterprises in other countries. In future research, it may be necessary to consider measuring objective performance rather than subjective performance as one way to solve these issues.

Line 450: Insert sentences

  • The failure to elucidate the impact of entrepreneurship on performance is considered a significant limitation of this study. In regression analysis with only entrepreneurship input, a significant positive relationship was observed. However, in regression analysis with all three leadership styles included, the relationship between entrepreneurship and performance was not significant. One possible explanation is that the impact of other leadership styles on performance is significant, which may have offset the effect of entrepreneurship on performance. However, further research is needed to verify this.

Comments # The leadership classification used, enshrined in the hypotheses, is not the only one. There are many approaches to this issue. Therefore, the authors need additional arguments in favor of choosing these three types of leadership for analysis.

  • Thanks for your accurate comments, again. I also completely agree that there is a need to add other leadership in addition to the three leaderships considered in this study. I further elaborated on this matter in the limitations section.

Line 444: Insert sentences

  • Another limitation of this study is the failure to provide clearer evidence regarding what constitutes the primary leadership in commercial enterprises. If this study had provided reasons why the three leadership styles considered here, rather than recently emphasized leadership such as authentic leadership, are more important or the evidence claiming them as the main leadership, the implications of this study could have been better elucidated.

Line 457: Insert sentences

  • Additionally, there is a need to study the relationship between leadership and performance by attempting a new approach to leadership effectiveness, such as leadership learning [48] or networking [49].

Comments # The article was submitted to the journal Sustainability, and therefore should cover sustainability issues. The authors are recommended to supplement the text of the Introduction and other sections of the article with a connection to sustainability. This will also require some changes to the title of the article.

Title Change

Does the Leadership of Commercial Enterprises also Contribute to the Improvement of Social Enterprises’ Outcomes?

  • The relationship between leadership and performance for enhancing the sustainability of social enterprises

 Line 7: Insert sentences

=> As social enterprises are established for the purpose of solving local problems, the sustainability of social enterprises is also important for local development. In order to increase the sustainability of social enterprises, performance improvement is necessary, and research on finding leadership suitable for social enterprises continues as an important method for performance improvement.

Sincerely.

Round 2

Reviewer 1 Report

Comments and Suggestions for Authors

The authors analyzed and improved the article taking into account the opinion/suggestions presented. It is only considered that some of the citations made, being a little old, should be cited in the text differently. Of course, it is important to contextualize the sample, but if the data used is relevant, it must be included in the discussion. The author chose to omit this data, table 1, which seems appropriate, given the discussion presented.

Reviewer 2 Report

Comments and Suggestions for Authors

Dear authors, 

After reviewing the proposed changes I consider that the concerns raised in the review have been successfully addressed. Therefore, I consider that the article has improved from its original version. Congratulations.

Best regards.